

# Stoichiometrically coupled carbon and nitrogen cycling in the MIcrobial-MIneral Carbon Stabilization model (MIMICS-CN)

Emily Kyker-Snowman[1], William R. Wieder[2,3], Serita Frey[1], A. Stuart Grandy[1]

[1]Department of Natural Resources and the Environment, University of New Hampshire, Durham, NH, USA
[2]Climate and Global Dynamics Laboratory, National Center for Atmospheric Research, Boulder, CO, USA
[3]Institute of Arctic and Alpine Research, University of Colorado, Boulder, CO, USA

*Correspondence to*: Emily Kyker-Snowman (ek2002@wildcats.unh.edu)

**Abstract.** Explicit consideration of microbial physiology in soil biogeochemical models that represent coupled carbon-nitrogen dynamics presents opportunities to deepen understanding of ecosystem responses to environmental change. The

MIcrobial-MIneral Carbon Stabilization (MIMICS) model explicitly represents microbial physiology and physicochemical stabilization of soil carbon (C) on regional and global scales. Here we present a new version of MIMICS with coupled C and nitrogen (N) cycling through litter, microbial, and soil organic matter (SOM) pools. The model was parameterized and validated against C and N data from the Long-Term Inter-site Decomposition Experiment Team (LIDET; 6 litter types, 10 years of observations, 13 sites across North America). The model simulates C and N losses from litterbags in the LIDET study

with reasonable accuracy (C: $R^2$=0.63, N: $R^2$=0.29) results that are comparable with simulations from the DAYCENT model that implicitly represents microbial activity (C: $R^2$=0.67, N: $R^2$=0.30). Subsequently, we evaluated equilibrium values of stocks (total soil C and N, microbial biomass C and N, inorganic N) and microbial process rates (soil heterotrophic respiration, N mineralization) simulated by MIMICS-CN across the 13 simulated LIDET sites against published observations from other continent-wide datasets. We found that MIMICS-CN produces equilibrium values in line with measured values, showing that

the model generates plausible estimates of ecosystem soil biogeochemical dynamics across continental-scale gradients. MIMICS-CN provides a platform for coupling C and N projections in a microbial-explicit model but experiments still need to identify the physiological and stoichiometric characteristics of soil microbes, especially under environmental change scenarios.

## 1 Introduction

Soils contain the largest actively cycling terrestrial carbon (C) stocks on earth and also serve as the dominant source of

nutrients, like nitrogen (N), that are critical for maintaining ecosystem productivity (Gruber and Galloway, 2008; Jobbágy and Jackson, 2000). Soil C cycle projections and their response to global change factors remain highly uncertain (Bradford et al., 2016; Todd-Brown et al., 2013), but recent empirical insights into microbial processing of soil C provide opportunities to update models and reduce this uncertainty (Cotrufo et al., 2013; Kallenbach et al., 2016; Lehmann and Kleber, 2015; Schmidt et al., 2011; Six et al., 2006). Several models have been developed recently with explicit representation of nonlinear microbial

C processing dynamics, including the MIcrobial-MIneral Carbon Stabilization (MIMICS) model (Sulman et al., 2018; Wieder





et al., 2014a, 2015b) and others (Abramoff et al., 2017; Allison, 2014; Fatichi et al., 2019; Hararuk et al., 2015; Robertson et al., 2018; Sulman et al., 2014; Wang et al., 2013, 2014a, 2017). These models are as good as or better than models without explicit microbial pools at simulating global soil C stocks and the response of soil C to environmental perturbations (Wieder et al., 2013, 2015b), and they also predict very different long-term responses of soil C to global change (Wieder et al., 2013,

2018). Microbial-explicit models have thus furthered our understanding of C cycling in the terrestrial system, but they also provide new opportunities to explore couplings between C and nutrient cycles, especially N.

Terrestrial models that couple C and N cycles reveal important ecosystem feedbacks that are absent from C-only models. For example, across ecosystems, experimental manipulations consistently indicate that N availability limits plant productivity (LeBauer and Treseder, 2008).  C-only model configurations in models typically predict that $CO_2$ fertilization

will result in a large increase in both plant productivity and the land C sink in coming decades, but nutrient limitation may constrain the magnitude of this terrestrial ecosystem C uptake  (Wieder et al., 2015a; Zaehle et al., 2015; Zaehle and Dalmonech, 2011). As terrestrial models increasingly represent coupled C-N biogeochemistry, accurate model estimates of N release from soil organic matter (SOM) will become important to reducing uncertainty in the $CO_2$ fertilization response of the terrestrial C cycle.

Currently, most biogeochemical models that couple C and N cycles have an implicit representation of microbial activity. These conventional models represent SOM decomposition with the assumption that chemical recalcitrance of organic matter dictates the turnover of litter and SOM pools (Luo et al., 2016). Carbon and N fluxes represented in these models are directly proportional to donor pool sizes, without any explicit representation of the microbes that mediate these fluxes (Schimel, 2001, 2013). Linear decay constants and transfer coefficients determine the flow of C and N through a decomposition cascade,

and rates of N immobilization and mineralization emerge from the interaction of fixed respiration fractions and the stoichiometry of donor and receiver SOM pools. The lack of plant-microbe-soil feedbacks in these models may limit their predictive capacity, especially in the face of environmental change. For example, in these models increased plant inputs to soil only build soil C and N stocks, and plants have no way to stimulate the microbial community to mine existing SOM for N without model modifications (Guenet et al., 2016; Wutzler and Reichstein, 2013). This "N mining" or "priming" effect, where

increased plant inputs result in increased microbial activity and decomposition rates, has been demonstrated in experimental studies (Cheng and Kuzyakov, 2005; Dijkstra et al., 2013; Phillips et al., 2012) and may be a critical pathway for plants to obtain more N and support increased plant productivity under elevated $CO_2$ (Thomas et al., 2015; Zaehle et al., 2014).

Microbes are critical mediators of soil C-N couplings and the release of plant-available N. As such, models that explicitly consider microbial activity provide an opportunity to explore potential microbial control over soil C-N

biogeochemical cycling and improve simulations of patterns in ecosystem C and N. Towards this end, multiple models have been introduced that explicitly consider the role of microbial activity in ecosystem C-N interactions  (Averill and Waring, 2017; Fatichi et al., 2019; Huang et al., 2018; Schimel and Weintraub, 2003; Sistla et al., 2014; Sulman et al., 2014, 2017, 2018, 2019; Wang et al., 2014a, 2017, 2013). To date, the majority of these microbial-explicit C-N models have been developed to explore soil biogeochemical interactions and microbial community dynamics, while only one has been validated for N





dynamics across a continental-scale gradient (Fatichi et al., 2019). Although there is great value in exploring diverse approaches to explicitly representing microbes in purely theoretical or site-specific applications, implementing these conceptual developments within larger-scale models requires convincing evidence that adding them improves model performance against large-scale data. Recent soil model comparisons report divergent responses to simulated global change experiments among microbial-explicit model formulations, highlighting the large uncertainty in their underlying process-level

representation and parameterization (Sulman et al., 2018; Wieder et al., 2018). The addition of explicit microbial pools may improve the predictive ability of landscape-scale models in the long run, but microbial models must be validated against landscape-scale datasets of a variety of pools and process rates before they can reasonably be expected to improve model performance and reduce uncertainty.

Given the value of coupled C-N microbial-explicit soil models for exploring mechanisms and the need for such

models to be validated across broad spatial scales (Louis et al., 2016), we developed a coupled C-N version of MIMICS. The C-only iteration of MIMICS considers trade-offs involved with microbial functional traits as well as both physicochemical (i.e. mineral associations) and chemical (i.e. recalcitrance) mechanisms of C stabilization in soil. Wieder et al. (2014, 2015b) and Sulman et al. (2018) evaluated this C only version of MIMICS across site, continental, and global scales. Here we expand on this work, introducing MIMICS-CN, which incorporates stoichiometrically coupled C and N cycling of all microbial, litter

and SOM pools and stoichiometric constraints on microbial growth. Our core objectives were to: 1) Formulate a framework and parameterization for coupled C and N cycling in MIMICS; 2) Validate MIMICS-CN against a continental-scale litter decomposition dataset (LIDET) and compare MIMICS-CN to a microbially-implicit, linear model (DAYCENT); and 3) Evaluate equilibrium soil and microbial stocks and fluxes (and their parameter sensitivities) that are simulated by MIMICS-CN with data synthesized across published landscape-scale data.

## 2 Methods

### 2.1 Model formulation

MIMICS-CN builds upon the previous C-only version of MIMICS, described in Wieder et al. (2014, 2015b). The C-only version of the model represents C flows through seven pools (Fig. 1): two litter pools, two microbial pools, and three SOM pools. Litter inputs to the model are partitioned into structural litter ($LIT_s$) and metabolic litter ($LIT_m$) pools based on measured

N and lignin in litter at a given site. Temperature-sensitive forward Michaelis-Menten kinetics determine the flux of litter pool C into rapidly-growing, r-strategist microbial biomass ($MIC_r$) and slower-growing, K-strategist microbial biomass ($MIC_K$). Fluxes of C into microbial pools result in respiration losses according to a defined carbon use efficiency (CUE) that varies by microbial functional group and substrate quality (e.g. structural or metabolic litter). Microbial pool sizes are moderated by inputs, CUE, and biomass-specific turnover rates. Microbial biomass turns over into physicochemically-stabilized ($SOM_p$) and

chemically-stabilized ($SOM_c$) soil organic matter pools. Desorption from $SOM_p$ and oxidation of $SOM_c$ feed into a pool of 'available' soil organic matter ($SOM_a$), which microbes can access via forward Michaelis-Menten kinetics. We implemented



density-dependent microbial turnover (sensu Georgiou et al., 2017; see Appendix A) for this iteration of the model to make microbial pools behave realistically in response to small changes in C inputs (Wang et al., 2014b, 2016). The density-dependent turnover of microbial biomass dampens the oscillatory response of microbial biomass to perturbations.

The current representation of N cycling in MIMICS-CN is based on the threshold element ratio idea described in Sinsabaugh et al. (2009) and Mooshammer et al. (2014) whereby organisms maintain biomass stoichiometry by spilling excess C or N on either side of a threshold ratio. We modified the C-only iteration of MIMICS to include N by adding a parallel set of pools and fluxes for N, as well as a pool for inorganic N (Fig. 1). The C cycle drives decomposition with fluxes from litter and SOM pools to microbes based on biomass-C-based forward Michaelis-Menten kinetics. Parallel N fluxes are determined

by the C:N ratio of the donor pools, which is a fixed parameter for the metabolic litter pool, varies with litter input chemistry for the structural litter pool, and depends on inputs for SOM pools. We use a fixed C:N of 15 for metabolic litter inputs, while the C:N of structural litter was allowed to vary to ensure conservation of total N inputs from litterfall (Table 1).

    The coupling between C and N cycles in MIMICS-CN occurs in the microbial biomass: at each hourly time step, the total C and N in incoming fluxes available to microbes is summed and adjusted based on the C use efficiency (CUE; varies

with microbial functional group and substrate) and N use efficiency (NUE; set to 0.85 for all fluxes entering microbial biomass pools in this model iteration). If the C:N of substrates being assimilated by microbial functional groups is greater or less than the C:N of the microbial biomass (defined as 6 and 10 for r- and K-strategists, respectively; Table 1), the microbes will spill excess C or N to maintain their biomass stoichiometry through overflow respiration or excess N mineralization. In MIMICS-CN the C:N ratio of SOM pools is flexible and determined by the inputs from microbial residues and direct inputs from litterfall

fluxes ($f_i$; Fig. 1). All N fluxes into microbial pools leak a small quantity of N into a dissolved inorganic N pool (DIN) based on the model-defined NUE. At each time step, each microbial functional group can access a fraction of the inorganic N pool proportional to their fraction of total microbial biomass. Plant N uptake and ecosystem losses (both hydraulic and gaseous) of inorganic N are handled implicitly at this stage, with a fixed fraction (20%) of DIN leaving the soil component model every time step.

**2.2 Model parameterization and validation: Cross-site litter decomposition**

We parameterized and validated MIMICS-CN using C and N dynamics observed across multiple sites participating in the 10-year Long-Term Intersite Decomposition Experiment Team (LIDET) experiment (Adair et al., 2008; Harmon et al., 2009; Parton et al., 2007). The LIDET study selected standardized plant litter types with a range of litter quality (lignin and N concentration), placed litterbags containing 100 g of each litter type at sites across a continental scale gradient of climatic

conditions, and measured changes in the C and N in litterbags on an approximately annual basis for 10 years. Although the original dataset included 27 sites across North America, we utilized data from 14 sites ranging from Alaska to Puerto Rico based on the data available at those sites to drive MIMICS (see Wieder et al., 2015b for site information). The LIDET dataset is a robust appraisal of the impacts of climate and litter chemistry on litter decomposition and has been used as a dataset for





comparing models of soil and litter decomposition in the past (Bonan et al., 2013). MIMICS has been used previously to
simulate C losses in the LIDET study (Wieder et al., 2015b).

We parameterized MIMICS-CN using observations from Harvard Forest in Petersham, MA, USA. Observations included both litterbag C loss and N data from the LIDET study as well as measurements of soil C and N stocks and microbial C and N from other studies at Harvard Forest (Colman and Schimel, 2013). Multiple combinations of parameters produced equally good fits to litter decomposition data; thus ancillary data on soil and microbial C stocks were used to inform the
parameter values presented here (Table 1). These ancillary data were not reported in LIDET and were not measured on identical plots to those used for the LIDET study (Harvard Forest encompasses multiple experiments and ecotypes), but these general targets were useful in distinguishing among model parameterizations. Our general targets for stocks at Harvard Forest included soil C and N (0-5 cm mineral soils, coniferous stand): 61 mg C cm$^{-3}$ and 2.9 mg N cm$^{-3}$; soil C:N: 21; and microbial biomass: 0.61 mg C cm$^{-3}$ (1% of soil C based on Xu et al. 2013).

After parameterizing the model to match observations at Harvard Forest, the model was validated using data from the remaining LIDET sites. To represent litterbags in MIMICS-CN, we first spun up the underlying model to simulate steady-state soil C and N pools and fluxes across sites in the LIDET study using site-level measurements of mean annual temperature, clay content, and litter input quantity, lignin content, and C:N (Wieder et al., 2015b). Then, we added a pulse of metabolic and structural litter based on the type of litter in the simulated litterbag. We tracked the C and N across all model pools for 10 years
and calculated the C and N in litterbags as the difference between total model C and N in the simulations and total model C and N at steady state. For each site, the model was sampled at time points equivalent to the real data collection dates in LIDET (approximately annually). Observed and modeled values of C and N in litterbags were compared by calculating R$^2$, root mean square error (RMSE) and bias.

To contextualize our results, we compared MIMICS-CN simulations of LIDET data against DAYCENT (Bonan et
al., 2013) simulations of the same data. Bonan et al. (2013) used the full complement of 27 LIDET sites in their analysis, but here we subset those results for the 13 sites used in the MIMICS-CN validation. We calculated R$^2$, RMSE and bias in the same way for each model and compared results across models, grouping results by biome.

**2.3 Model evaluation: Equilibrium C and N cycling**

Building on the LIDET simulations, we independently synthesized observations to evaluate the patterns of C and N pools and
fluxes across a variety of sites. Although direct, site-specific comparisons of modeled and observed values like microbial biomass would have been ideal, MIMICS-CN represents many variables that were not measured in the LIDET study and have not been synthesized across these Long-Term Ecological Research sites. Instead, we compared the range and distribution of pools (soil organic C and N, microbial biomass C and N, and total inorganic N) and fluxes (heterotrophic respiration and N mineralization) using the modeled LIDET simulations and published syntheses of observations from other sites (Cleveland
and Liptzin, 2007; Colman and Schimel, 2013; Xu et al., 2013; Zak et al., 1994). To more directly compare measurements with model results, stock measurements were converted to units of % of soil mass and fluxes (heterotrophic respiration and





net N mineralization rates) were converted to units of µg cm$^{-3}$ hr$^{-1}$. MIMICS reports pool values in units of g cm$^{-2}$ (0-30 cm); to compare MIMICS against observations we converted MIMICS values to % by mass assuming a bulk density of 1.5 g cm$^{-2}$. Soil depth simulated by MIMICS (30 cm) is deeper than most of the observations in the compiled dataset, but the purpose of

this exercise was to evaluate whether MIMICS produces realistic values for soil biogeochemical stocks and fluxes across continental-scale ecoclimatological and edaphic gradients, rather than making a direct site-specific comparison. The distribution of values produced by MIMICS across the LIDET sites was superimposed on the distributions of observed values to illustrate data-model agreement and to visualize the median and range of measurements across studies.

Finally, we documented relationships between model input variables (mean annual temperature, productivity, clay

content, and litter quality) and the distribution of SOM pools that were simulated at the LIDET sites. Our aim with these analyses was to illustrate the underlying assumptions in the model and how they influence the size and distribution of C across SOM pools. The SOM pools represented in MIMICS have different stabilization mechanisms (chemical or physicochemical) operating in parallel, as opposed to a cascade of successively more recalcitrant pools as simulated in first-order models. We wanted to explore how assumptions made in the model structure and parameterization of MIMICS determine the quantity and

distribution of SOM pools, and how they change among sites with variation in climatic, biological, and edaphic properties. To do this we looked at the absolute and relative contributions of each SOM pool simulated by MIMICS across the LIDET sites and conducted linear regressions to determine how environmental factors control their distributions. We also conducted linear regressions between soil C:N and both litter chemistry and environmental factors to assess the drivers of soil C:N in the model.

## 3 Results

### 3.1 Model parameterization and validation: Cross-site litter decomposition

We parameterized MIMICS-CN to replicate litter C decay rates and N dynamics of six litter types observed in the LIDET study at the Harvard Forest LTER site (Fig. 2). In its current parameterization, MIMICS slightly overestimates litter C loss at later stages of decay, but most time points are within uncertainty estimates of the observations (Fig. 2a). Similarly, for N, MIMICS-CN overestimates N accumulation in early stages of decay and underestimates N remaining at later stages, but most

time points follow a reasonable trajectory given observations. MIMICS-CN also captures the effects of litter quality on both rates of litter decay (Fig. 2a) and litterbag N accumulation (Fig. 2b). The parameters we used to fit MIMICS-CN to Harvard Forest data also produce reasonable estimates of soil N stocks (2.0 vs. 2.9 mg N cm$^{-3}$ for model and observations, respectively) and microbial biomass (0.65 vs 0.61 mg C cm$^{-3}$), although estimates of soil C (21 vs 61 mg C cm$^{-3}$) and soil C:N (11 vs. 21) are both lower than observations.

Parameter values used for this and subsequent simulations across all LIDET sites are shown in Table 1. Relative to the previous C-only version of the model (Wieder et al., 2014a, 2015b), kinetic parameters and microbial turnover values were adjusted to account for density-dependent turnover (Georgiou et al. 2017). In addition, the fraction of structural litter that bypasses microbial biomass to enter the chemically-protected pool ($f_i$) was increased from 5% to 30% as a means to produce





reasonable values for total soil C:N. Finally, we adjusted the partitioning of microbial turnover to stable soil pools in order to
more closely match distributions at Harvard Forest.

Applying this parameterization across all six litter types at 13 LIDET sites, MIMICS-CN simulates C losses and N dynamics from litterbags with an $R^2$ of 0.63 and 0.29 and an RMSE of 16.0 and 0.34, respectively. (Fig. 3). MIMICS-CN captures effects of litter quality on decay rates, with faster rates of C loss and more rapid N mineralization simulated with more N rich *Drypetes glauca* litter, and slower rates of C loss and greater N immobilization simulated by low quality *Triticum*
*aestivum* litter (Fig 3a, c). MIMICS-CN is best at capturing C loss rates in high- and intermediate-quality litters (*Drypetes glauca*, *Pinus elliottii, Thuja plicata*, and *Acer saccharinum*) but tends to underestimate litter C loss rates from the lowest-quality litter (*Triticum aestivum*). For N immobilization and loss, the model performs well especially for high-quality litters but underestimates N accumulation slightly in the lowest-quality litter. The model also captures broad climate effects on litter C loss, with slower decay rates in tundra and boreal forests sites and faster decay in tropical and deciduous forests (Fig 3b).

MIMICS-CN and DAYCENT simulations of LIDET decomposition data are compared in Table 2. Across a broad range of biomes, MIMICS-CN simulates litter decomposition dynamics as well as or better than DAYCENT. For C, MIMICS-CN produces lower RMSE values for some biomes (arid, tropical) but not others (tundra, boreal, conifer, deciduous and humid), although the values were generally very close between the two models. For N, MIMICS-CN produces a lower RMSE for conifer, deciduous, humid, arid, and tropical biomes, although the differences are slight. MIMICS-CN outperformed
DAYCENT in the warmest biomes while DAYCENT excelled for colder sites for both C and N, but the differences in model fit to data were slight and would be difficult to attribute to any particular differences in model structure. DAYCENT simulates decomposition based on initial litter chemistry and showed no site-specific effects on the maximum N immobilized or the relationship between C and N during decomposition for a given litter type (Fig. S1 and S2). By contrast, the amount of N that can be immobilized by a litterbag in MIMICS-CN is driven by the availability of N and the stocks and flows of N in the
simulated steady-state soil, and MIMICS-CN showed site-specific variability in the shape of N immobilization and loss curves (Fig. 3 and 4).

Litter quality determines the timing of N immobilization vs. mineralization in observations. This produces a functional relationship between initial litter chemistry, C loss, and N immobilization / mineralization that is fairly consistent across sites (colored dots; Fig. 4). MIMICS-CN broadly captured litter quality effects on the timing and magnitude of N
immobilization and mineralization dynamics across all biomes (red dots; Fig 4). For example, litter with high initial chemical quality consistently mineralize N throughout all stages of litter decay, and MIMIC-CN adequately captures this functional C-N relationship (Fig 4a,b). By contrast, litters with lower initial chemical quality immobilize N during early stages of litter decay, but subsequently mineralize N as decomposition proceeds. MIMICS-CN broadly captures these patterns, but without as much variation as the observations (Fig 4c-f). The lowest-quality litter (*Triticum aestivum*) immobilizes N until only 40%
of C remains in litterbags. Although MIMICS-CN potentially underestimates total N immobilization *Triticum aestivum* litter, it does capture the point at which net N mineralization begins (Fig. 4f).



## 3.2 Model evaluation: Equilibrium C and N cycling

Across all sites and litter types in the LIDET simulations, the ranges of underlying pool sizes and process rates in MIMICS-CN were compared against published ranges from similarly diverse sets of sites (Cleveland and Liptzin, 2007; Colman and

Schimel, 2013; Xu et al., 2013; Zak et al., 1994). MIMICS-CN simulations produced reasonable equilibrium values for most pools and fluxes (Table 3 and Fig. 5). In general, the range of values across the 13 sites simulated by MIMICS was smaller than the ranges across the thousands of sites included in the compiled dataset of observations. For example, total soil C ranged from 7.0-50 mg C cm$^{-3}$ in MIMICS simulations but ranged from 2.7-610 mg C cm$^{-3}$ in observations. Despite this discrepancy, the median values of the simulations and observations were generally within reason (Fig. 5). The distributions of measured

and modeled values for microbial biomass C and N as a percent of total soil C and N overlapped closely, providing confidence that the model reasonably represents microbial stoichiometry, microbial activity as a function of biomass, and microbial biomass as a function of SOM. For soil C:N, the model tended to produce low values relative to the range and median of observed values.

Finally, we explored the environmental controls on the distribution of SOM across physicochemically-protected,

chemically-protected, and available pools in MIMICS-CN by examining the correlations between pool sizes and salient input variables (mean annual temperature, productivity, clay content, and litter lignin content). The results are shown in Figure 6. The absolute concentration of SOM simulated across the LIDET sites was most strongly correlated with ANPP ($R^2$=0.52), but also tended to increase with MAT, albeit inconsistently (Fig. 6a; $R^2$=0.15). The distribution of SOM across stabilized pools strongly favored chemically-protected SOM at sites with lower temperatures, while the relative proportion of

physicochemically-protected SOM increased with increasing temperature (Fig. 6b). The relative proportion of SOM in the available pool remained fairly consistent across simulated sites. Physicochemically-protected SOM was tightly positively correlated with the product of ANPP and clay content ($R^2$=0.96, Fig. 6c), while chemically-protected and available SOM were negatively correlated with MAT (Fig. 6d, $R^2$=0.40 and 0.47, respectively) and positively correlated with litter lignin content (Fig. 6e; $R^2$=0.68 and 0.32, respectively). The C:N of individual pools was fairly consistent across sites and tended to be higher

for chemically-protected SOM (~15) than available (~8) or physicochemically-protected SOM (~10). As a result, soil C:N was largely driven across sites by the distribution of SOM across pools, especially the absolute size of the SOMp pool (Fig. 6f, $R^2$=0.79). Given that clay content was an important driver of physicochemically-protected SOM in the model, clay content was tightly correlated with soil C:N ($R^2$=0.88). Other litter characteristics and environmental factors were not strong drivers of soil C:N ($R^2$ for MAT: 0.42; litter lignin: 0.03; litter C:N: 0.005).

## 4 Discussion


Terrestrial models are increasingly representing coupled C-N biogeochemistry, and MIMICS-CN is among the first attempts to do so with a microbial explicit soil biogeochemical model that can be used to project C and N dynamics across continental-scale gradients. Our formulation and parameterization of MIMICS-CN captures site level observations of litter C loss and N



immobilization at the Harvard Forest LTER site (Fig. 2). Cross-site validation of the model demonstrates that it broadly
captures climate and litter quality effects on rates of C and N transformations from the LIDET observations (Figs. 3-4).
Notably, the results simulated by MIMICS-CN represent N dynamics during litter decomposition as well as or better than a
first-order model that implicitly represents microbial activity (Table 2). It also generates steady state pools and fluxes of C and
N that seem reasonable compared to published syntheses (Table 3; Fig. 5). Below we discuss these dynamic and equilibrium
model simulations in greater detail, as well as some of the limitations of MIMICS-CN that will be addressed in future work.

**4.1 Model parameterization and validation: Cross-site litter decomposition**

We first parameterized and validated MIMICS-CN using the cross-site litter decomposition study, LIDET. Previous LIDET
simulations using MIMICS have successfully replicated observed C loss patterns, and adding coupled N cycling to MIMICS
neither improved nor degraded simulations of LIDET litter C losses relative to the C-only model (Figs. 2-3; Wieder et al.
(2015b) report global RMSE for the C-only model = 14.6 vs. 16.0 in this study). This result is explained by the nature of
belowground C-N couplings in the new model: although C and N flow together through model pools, model dynamics are
primarily driven by C, with N dynamics following suit based on pool stoichiometry. The N dynamics only constrain C cycling
in the model if microbes are N-limited, in which case microbes lose excess C through overflow respiration. At equilibrium,
microbes in MIMICS-CN primarily obtain N through recycling of SOM pools with favorably low C:N ratios, with the result
that modeled microbes are almost always C-limited at equilibrium. Large pulses of low-quality litter can perturb this
equilibrium and induce N limitation, but in the absence of losses of or plant competition for inorganic and dissolved organic
N, C cycling in MIMICS proceeds in essentially the same way with or without accounting for N.

MIMICS-CN accurately captured the stoichiometric relationships between C and N during litter decomposition (Fig.
4). This stoichiometric relationship has been well-defined in the past using theoretical microbial stoichiometry and CUE
(Parton et al., 2007), but comparable soil models without explicit microbial physiology have tended to over-predict N
accumulation in litterbags (Bonan et al., 2013). Moreover, models without microbial explicit physiology also show N
immobilization mineralization dynamics that are completely determined by initial litter quality, whereas MIMICS simulations
show greater site-level variation (Fig 4; SI Figure S2). In MIMICS-CN, stoichiometric relationships drive litterbags to
accumulate soil N until they reach a threshold C:N, after which litterbags become net sources of N. This threshold, representing
the balance between microbial N requirements and availability, is a function of changes in litter stoichiometry during
decomposition, as well as of the stoichiometry of microbes and their nutrient use efficiencies. By explicitly considering these
dynamics MIMICS-CN has a similar or lower RMSE for N remaining in litter bags than a model that implicitly represents
microbes, DAYCENT (Table 2).

MIMICS-CN and DAYCENT capture N dynamics during decomposition with similar overall degrees of fit, but for
different reasons. In DAYCENT, N immobilization and loss dynamics are driven by initial litter chemistry, and good model
fit to data is achieved by capturing the average N immobilized for a given litter type regardless of biome and climate conditions
(see Fig. S1 and S2). By contrast, litterbag N immobilization in MIMICS-CN is driven by the availability of N in the underlying





modeled soil and by site-specific effects (e.g. climate, clay content) on the simulated stocks and fluxes of N. As a result, MIMICS-CN generates greater variation in the amount N immobilized for a given litter type across sites (Fig. 3 and 4). Site-specific variability in N immobilization patterns is also clearly visible in LIDET observations (colored dots, Fig. 4), but the

introduction of site-specific variability in MIMICS-CN does not substantially improve model fit to data relative to DAYCENT. Spatial variability in ecosystem processes, like N mineralization rates, may be linked to local-scale microbial community composition (Graham et al., 2016; Smithwick et al., 2005; Soranno et al., 2019). While more work needs to be done to understand the factors controlling within and among site variation in soil C-N dynamics (Bradford et al., 2017), these results highlight that the explicit representation of microbial activity in MIMICS-CN may present opportunities to explore factors

responsible for biogeochemical heterogeneity across scales.

Although MIMCS-CN broadly captures appropriate climate and litter quality effects on leaf litter decomposition patterns, the model underestimates N accumulation in the highest C:N ratio litter (*Triticum aestivum*; Fig. 4f). Microbes in MIMICS-CN recycle nitrogen from necromass and necromass-derived SOM, which might allow microbes to scavenge the N required to decompose high C:N litter without having to accumulate it from the inorganic soil pool. In a real litterbag,

necromass might be lost through leaching and microbial access to recycled biomass might be limited, and some microbial-derived compounds may require extensive depolymerization and proteolysis before the N is available for recycling (Schulten and Schnitzer, 1997), thus favoring N uptake from the soil pool. Nonetheless, the high C:N ratio of *Triticum aestivum* is not typical of the majority of litter inputs across diverse biomes (Brovkin et al., 2012) which are well within the range that MIMICS-CN can simulate.

**4.2 Model evaluation: Equilibrium C and N cycling**

We conducted additional model evaluation by comparing model pools and fluxes at equilibrium to published observations. The parameter values used in the LIDET simulations produced reasonable estimates of equilibrium pools (soil organic C and N, microbial biomass C and N, and total inorganic N) and fluxes (heterotrophic respiration and N mineralization) (Table 3; Fig. 5). In combination with the LIDET results, these results indicate that MIMICS-CN can produce realistic simulations of

both the short-term dynamic processes involved in litter decomposition and the soil-forming processes that produce equilibrium pools and fluxes over much longer time scales. In addition, these reasonable estimates emerge from the representation of microbial stoichiometry, microbial growth and turnover, and microbially-mediated decomposition in MIMICS-CN, rather than from prescribed values for soil C stocks or decomposition rates as in some conventional models that lack explicit representation of microbes. This increases the power of MIMICS-CN, relative to these other models, to explore

the microbial and biogeochemical processes underpinning model predictions.

Continent-wide observation of soil pools and fluxes range over several orders of magnitude (Table 3), but MIMICS simulations agreed well with the middle of those ranges. Observations tended to be spread over a much larger range of values than the MIMICS-CN simulations, but these simulations only included information from 13 sites while the observations included thousands of locations. The median values of observed and simulated values were within a factor of 2.5 for all pools





(Fig 5). Differences in measurement depth or error in estimated bulk density values could account for some of the differences between measurements and simulations and for the spread across observed values. This is less of a concern for three of the variables used here (soil C:N, microbial biomass C as a percent of total soil C and microbial biomass N as a percent of total soil N), which are ratios that are comparable across sites. Microbial biomass C as a percent of total soil C and microbial biomass N as a percent of total soil N were highly conserved across sites, relative to soil stocks or microbial C or N, and may

be particularly useful metrics for evaluating microbial explicit soil biogeochemical models since the size of the microbial biomass pool directly controls rates of SOM turnover and formation in models like MIMICS-CN. For these ratios, MIMICS-CN reproduced distributions and median values that overlapped well with observations. In future work, direct comparisons of modeled and measured values for these ratios at specific sites may shed light on the limitations of the model and the origins of data-model disagreement. However, even the simple range comparisons included here provide evidence that the mechanistic

representation of soil biogeochemistry in MIMICS-CN is ecologically realistic. Examinations of model realism like this are a crucial step in transitioning from theory and small-scale model tests to applications in ESMs or at larger scales where evaluation data are more sparse.

Besides representing appropriate soil biogeochemical stocks, fluxes simulated by the models also agree well with observations. Specifically, MIMICS-CN simulations of heterotrophic respiration and net N mineralization rates fell within

observed bounds, although the variation in observations was much greater than the variation in simulated values. Our simulations calculated rates at equilibrium assuming constant temperature and other factors, while real rates of these processes are driven by seasonally- and diurnally-variable temperature, soil moisture, and other factors, so predictably, our simulations produced smaller-than-observed variability in rates. MIMICS-CN produced total soil C:N values that fall within observed ranges, although observations again show greater variation of soil C:N ratios and have maximum values that are much higher

than the maximum C:N ratios simulated by MIMICS-CN. SOM pools in MIMICS-CN are mostly comprised of microbial necromass, in addition to a small proportion of litter that enters SOM pools directly without first passing through microbial biomass. Increasing this proportion in the model is one way to increase the C:N of SOM pools and the overall system at equilibrium. At some sites, litter may contribute more directly to SOM pools than microbial necromass (Jilling et al., 2018). For example, forests often have a higher proportion of total soil C in the light fraction, which is almost entirely made up of

plant residues,  compared to agroecosystems and many grasslands (Grandy and Robertson, 2007).  For those sites with large, direct contributions of plant matter to SOM, increasing the fraction of litter that passes directly into SOM in MIMICS may be appropriate.

The distribution of SOM across simulated pools in MIMICS-CN was driven by environmental variables in ways that make sense given the stabilization mechanisms defined for each pool. The chemically-protected and available SOM pools in

MIMICS-CN turn over based on temperature-sensitive, microbial-biomass-dependent Michaelis-Menten kinetics, and the flow of litter inputs into each microbial group and eventually into these pools is driven by litter chemistry. Therefore, these pools were negatively correlated with MAT and positively correlated with litter lignin content (Fig. 6d, 6e) across the simulated sites. Turnover of the physicochemically-protected SOM pool, on the other hand, occurs via first-order kinetics with a rate





constant modified by clay content, and the equilibrium values of this pool are determined by inputs that largely come from
microbial biomass (Fig. 1) and this turnover rate. Therefore, the equilibrium values of physicochemically-protected SOM were
positively correlated with the product of ANPP and clay content (Fig. 6c). Across the handful of sites included in these
simulations, chemically-protected SOM formed a higher proportion of total SOM at lower MAT, while physicochemically-
protected SOM was favored at warmer sites (Fig. 6b). Total simulated SOM in MIMICS-CN was driven by the combination
of factors that drive the individual pools; thus, total SOM was correlated with ANPP, MAT, and clay content.

365         Soil C:N ratios simulated by MIMICS-CN across sites were highly correlated with soil clay content ($R^2$=0.88),
suggesting that, in the model, soil stoichiometry emerges from the relative contributions of SOM across physicochemically-
and chemically-protected pools (Figure 6). Although the spread of C:N values across the sites simulated by MIMICS-CN was
small (Fig 6f), C:N tended to decrease with increasing temperature, and simulated soil C:N was more correlated with site
temperature ($R^2$=0.42) than any of the litter characteristics used to drive the model, such as litter lignin ($R^2$=0.03) or litter C:N
($R^2$=0.005). The lack of correlation between simulated soil C:N and litter C:N suggests an intriguing follow-up question: in
the field, is SOM stoichiometry correlated with litter quality, or is it better explained by climate and edaphic gradients that
impact soil microbial community composition, microbial activity, and mechanisms of SOM persistence? Presently, MIMICS-
CN assumes that microbial biomass stoichiometry largely controls soil C:N ratios, with relatively minor contributions from
litter quality. However, a small proportion of litter inputs become stabilized in MIMICS-CN without first passing through the
stoichiometric filter of microbial biomass, and increasing this fraction in the model is a means to increase the C:N of stable
SOM in the model. This result implies that in the field, C:N stoichiometry might be used as a means to differentiate the degree
to which a given soil fraction is derived from direct plant inputs or microbial biomass. Future work will use measured C:N of
soils and soil fractions and isotopic insights into the plant or microbial origins of stable SOM to improve the parameterization
of this aspect of the model and better understand the relationship between mechanisms of SOM stabilization and soil
stoichiometry.

### 4.3 Known limitations of MIMICS-CN and future work

MIMICS-CN combines reasonable biogeochemical simulations with the option to explore underlying microbial processes, but
limitations remain. For example, the stoichiometric coupling of C and N in MIMICS may diverge from reality under a number
of circumstances. The stoichiometric theory applied in biogeochemical models like MIMICS-CN relies on simplified
parameterizations for CUE, NUE, and microbial C:N, which in reality encapsulate an enormous number of different and
competing metabolic pathways (Sinsabaugh and Shah, 2012). For example, CUE is defined in MIMICS-CN as the ratio of
biomass C growth to substrate uptake, but in reality represents the balance of diverse catabolic (energy-producing) and anabolic
(biomass-producing) metabolic pathways that may shift in magnitude depending on the type of biomass a microbe is producing
(e.g. fungal hyphae vs. spores vs. energy storage compounds vs. enzymes). Moreover, the functional relationships between
stoichiometric parameters and substrate or environmental drivers are still murky (Geyer et al., 2016). CUE and NUE are critical
parameters in MIMICS-CN that determine whether microbes are net N mineralizers or immobilizers under any given substrate



conditions, but the relationships between CUE and temperature (Allison, 2014; Dijkstra et al., 2011; Frey et al., 2013; Steinweg et al., 2008), substrate quality (Blagodatskaya et al., 2014; Frey et al., 2013; Sinsabaugh et al., 2013), microbial community composition (Maynard et al., 2017), microbial growth rate (Molenaar et al., 2009; Pfeiffer et al., 2001), or any number of other

aspects of microbial metabolism are complex, difficult to quantify, and challenging to represent at the scale of a whole soil community. Microbial C:N is also assumed to be fixed for each microbial group in MIMICS, and while there is some support for narrow ranges for microbial C:N (Cleveland and Liptzin, 2007; Kallenbach and Grandy, 2011; Xu et al., 2013), microbial C:N may be somewhat flexible in reality. MIMICS only represents two microbial groups with different stoichiometric parameters, but real soils contain a much more diverse array of microbial functional groups with different responses to

environmental conditions and different couplings between C and N cycles. Despite these limitations, MIMICS and microbial models like it are a good first step towards representing the complex ecological factors that drive the coupling of soil C and N biogeochemistry. Future work could compare model formulations that take different approaches to microbial community and stoichiometric parameters (e.g. flexible parameters, additional microbial groups, partitioning microbial metabolism into a greater number of pathways) and assess the ramifications of different choices for simulating existing data and predicting the

long-term response of soil C and N cycles to global change.

       In its current configuration, MIMICS-CN simplifies a number of ecosystem biogeochemical processes, and there are several important pathways of N cycling currently absent from the model. For example, MIMICS-CN does not currently represent free living biological N fixation, direct mycorrhizal exchanges for plant C for microbial N, dissolved organic C or N losses, denitrification/nitrification/other inorganic N transformation and loss pathways, plant uptake of N, or inorganic N

leaching beyond a simple linear decay rate. Some of these shortcomings may be remedied by integrating MIMICS with a full ecosystem biogeochemical model that represents the greater complexity of the plant-soil continuum. Given the differences in soil C dynamics that are simulated by models that explicitly represent microbial activity (Wieder et al.2018; Sulman et al 2018), we expect to see similar differences in simulated N dynamics with MIMICS-CN (see also Sulman et al. 2019). For example, $CO_2$ fertilization is expected to increase plant productivity and inputs to soil, and first-order linear models of soil

predict increases in soil C with increasing inputs, while microbial models show more muted gains in soil C because increased inputs stimulate microbial activity (Wieder et al., 2015b). Under the same scenario, MIMICS-CN might also show increases in N released from stable pools with increasing inputs as a result of increased microbial activity. In a simulation coupled to a full terrestrial biogeochemical model, this might result in a feedback loop that would increase plant productivity and inputs, given that supplies of N are expected to limit the plant productivity response to $CO_2$ fertilization (Thomas et al., 2015; Zaehle

et al., 2014). The interactions between explicit microbial representation, coupled C-N biogeochemistry, and global change factors are likely to be complex in models. Models like MIMICS-CN provide a pathway to reconcile mechanistic explanations for phenomena like priming and plant-soil feedbacks with emergent patterns in terrestrial biogeochemistry across landscapes.



## 4.4 Conclusions

MIMICS-CN is one of the first models to integrate modern theories of SOM stabilization with explicit representation of
microbial C and N processing. As such, it provides a powerful opportunity to explore relationships among ecosystem
properties, climate drivers, and microbial community impacts on C and N cycling. MIMICS-CN integrates theories about
temperature-sensitive litter decomposition, microbial growth and death, and SOM stabilization. Future model development
will focus on utilizing large-scale datasets to constrain each of these components. If the theories represented in MIMICS prove
capable of replicating large-scale patterns in soil C and N cycling, it could go a long way towards improving confidence in
simulated terrestrial C cycle-climate feedbacks (Bradford et al., 2016). Our work demonstrates that a microbially-explicit
biogeochemical model can reproduce site and litter quality effects on litter decomposition C and N dynamics at a landscape
scale. Others have made the case that microbes *should* be incorporated into models (Todd-Brown et al., 2012), but *how* they
are incorporated into models matters (Luo et al., 2016). Moving forward, we need more data to make better-informed decisions
about the representation of microbes in models. The sign and magnitude of the response of microbial-explicit soil models to
global change relies on a wide range of choices about the structure and parameters used in the model, as well as the way that
field data is interpreted within the model. Cataloging these choices and their impacts helps us better approximate the truth and
develop soil models that will reduce uncertainty in larger models rather than just increasing their complexity.

## Code and data availability

MIMICS-CN is written in R using packages rootSolve (Soetaert and Herman, 2009) and hydroGOF (Zambrano-Bigiarini,
2017). Figures were generated using packages ggplot2 (Wickham, 2016), reshape2 (Wickham, 2007), scales (Wickham,
2018), gridextra (Auguie, 2017), and cowplot (Wilke, 2016). The R scripts and datasets used to generate model results are
available at https://zenodo.org/record/3534562. See Appendix A for equations.

## Competing interests

The authors declare that they have no competing interests.

## 445  Author contributions

E. Kyker-Snowman developed new model code and conducted model parameterization and testing with feedback from W. R.
Wieder and A. S. Grandy. W. R. Wieder developed the code for the original C-only MIMICS model. A. S. Grandy supervised
model development and testing. S. Frey provided advice on Harvard Forest data used to parameterize and evaluate the model
and contributed intellectually during manuscript development. E. Kyker-Snowman prepared the manuscript with contributions
from all co-authors.





**Acknowledgements**

Funding for this study was provided by the USDA National Institute of Food and Agriculture (Project No. 2015-35615-22747) and the US Department of Energy (Grant Number DE-SC0016590). E. Kyker-Snowman was supported by an NSF Graduate Research Fellowship under Grant No. DGE-1450271. Partial funding was provided by the New Hampshire Agricultural Experiment Station. W. Wieder was supported by grants from US Department of Energy, Office of Science, Biological and Environmental Research (BER) under award numbers TES DE-SC0014374 and BSS DE-SC0016364 and the USDA National Institute of Food and Agriculture 2015-67003- 23485.


**Appendix A: Model equations**

The structure and assumptions in the C-only version of MIMICS have been described previously (Wieder et al., 2014b, 2015b),
and the structure and assumptions in MIMIC-CN are described in section 2.1 ("Model formulation") of the methods section of this paper. The C fluxes (mg C cm$^{-3}$ h$^{-1}$) from donor to receiver pools in MIMICS-CN, numbered on Fig. 1, are defined by the following:

$$LIT_{m,C}\_MIC_{r,C} = MIC_{r,C} \times V_{max[r1]} \times LIT_{m,C} / (K_{m[r1]} + LIT_{m,C}), \tag{A1}$$

$$LIT_{s,C}\_MIC_{r,C} = MIC_{r,C} \times V_{max[r2]} \times LIT_{s,C} / (K_{m[r2]} + LIT_{s,C}), \tag{A2}$$

$$SOM_{a,C}\_MIC_{r,C} = MIC_{r,C} \times V_{max[r3]} \times SOM_{a,C} / (K_{m[r3]} + SOM_{a,C}), \tag{A3}$$

$$MIC_{r,C}\_SOM_C = MIC_{r,C}{}^{\beta} \times \tau_{[r]}, \tag{A4}$$

$$LIT_{m,C}\_MIC_{K,C} = MIC_{K,C} \times V_{max[K1]} \times LIT_{m,C} / (K_{m[K1]} + LIT_{m,C}), \tag{A5}$$

$$LIT_{s,C}\_MIC_{K,C} = MIC_{K,C} \times V_{max[K2]} \times LIT_{s,C} / (K_{m[K2]} + LIT_{s,C}), \tag{A6}$$

$$SOM_{a,C}\_MIC_{K,C} = MIC_{K,C} \times V_{max[K3]} \times SOM_{a,C} / (K_{m[K3]} + SOM_{a,C}), \tag{A7}$$

$$MIC_{K,C}\_SOM_C = MIC_{K,C}{}^{\beta} \times \tau_{[K]}, \tag{A8}$$

$$SOM_{p,C}\_SOM_{a,C} = SOM_{p,C} \times D, \tag{A9}$$

$$SOM_{c,C}\_SOM_{a,C} = (MIC_{r,C} \times V_{max[r2]} \times SOM_{c,C} / (KO_{[r]} \times K_{m[r2]} + SOM_{c,C})) \; +$$
$$\quad (MIC_{K,C} \times V_{max[K2]} \times SOM_{c,C} / (KO_{[K]} \times K_{m[K2]} + SOM_{c,C})). \tag{A10}$$


where pools and parameters are described in section 2.1 and Table 1, respectively. The N fluxes (mg N cm$^{-3}$ h$^{-1}$) from donor to receiver pools in MIMICS-CN are calculated based on the C fluxes between pools and the C:N ratio of donor pools. These fluxes are numbered on Fig. 1 and defined by the following:

$$LIT_{m,N}\_MIC_{r,N} = A1 \times LIT_{m,N} / LIT_{m,C}, \tag{A11}$$

$$LIT_{s,N}\_MIC_{r,N} = A2 \times LIT_{s,N} / LIT_{s,C}, \tag{A12}$$





$$SOM_{a,N}\_MIC_{r,N} = A3 \times SOM_{a,N} / SOM_{a,C}, \qquad (A13)$$

$$MIC_{r,N}\_SOM_{,N} = A4 \times MIC_{r,N} / MIC_{r,C}, \qquad (A14)$$

$$LIT_{m,N}\_MIC_{K,N} = A5 \times LIT_{m,N} / LIT_{m,C}, \qquad (A15)$$

$\quad LIT_{s,N}\_MIC_{K,N} = A6 \times LIT_{s,N} / LIT_{s,C}, \qquad (A16)$

$$SOM_{a,N}\_MIC_{K,N} = A7 \times SOM_{a,N} / SOM_{a,C}, \qquad (A17)$$

$$MIC_{K,N}\_SOM_{,N} = A8 \times MIC_{K,N} / MIC_{K,C}, \qquad (A18)$$

$$SOM_{p,N}\_SOM_{a,N} = A9 \times SOM_{p,N} / SOM_{p,C}, \qquad (A19)$$

$$SOM_{c,N}\_SOM_{a,N} = A10 \times SOM_{c,N} / SOM_{c,C}. \qquad (A20)$$


Each time step, the microbial pools in MIMICS-CN take up inorganic N from the DIN pool proportional to the biomass in each pool. Subsequently, the C:N ratio of all the inputs to each microbial pool is calculated, and the microbial pools spill either excess C or excess N to maintain a model-defined C:N ratio of microbial biomass. The algorithm that determines the release of excess C or N is determined using the following equations:


$$DINup_r = (1 - N_{leak}) \times DIN \times MIC_{r,C} / (MIC_{r,C} + MIC_{K,C}), \qquad (A21)$$

$$DINup_K = (1 - N_{leak}) \times DIN \times MIC_{K,C} / (MIC_{r,C} + MIC_{K,C}), \qquad (A22)$$

$$upMIC_{r,C} = CUE_{[1]} \times (A1 + A3) + CUE_{[2]} \times (A2), \qquad (A23)$$

$$upMIC_{r,N} = NUE \times (A11 + A13 + A12) + A21, \qquad (A24)$$

$\quad CNup_r = A23 / A24, \qquad (A25)$

$$Overflow_r = A23 - (A24 \times \min(CN_r, A25)), \qquad (A26)$$

$$Nspill_r = A24 - (A23 / \max(CN_r, A25)), \qquad (A27)$$

$$upMIC_{K,C} = CUE_{[3]} \times (A5 + A7) + CUE_{[4]} \times (A6), \qquad (A28)$$

$$upMIC_{K,N} = NUE \times (A15 + A17 + A16) + A22, \qquad (A29)$$

$\quad CNup_K = A28 / A29, \qquad (A30)$

$$Overflow_K = A28 - (A29 \times \min(CN_K, A30)), \qquad (A31)$$

$$Nspill_K = A29 - (A28 / \max(CN_K, A30)). \qquad (A32)$$

Inorganic N leaches slowly from the model according to a model-defined rate:


$$LeachingLoss = N_{leak} \times DIN. \qquad (A33)$$

Given the fluxes defined above, the changes in C and N pools in each hourly timestep (mg C or N cm$^{-3}$) are described by the following:




$$\frac{dLIT_{m,C}}{dt} = I_{LIT_{m,C}} \times \left(1 - f_{i,met}\right) - A1 - A5, \tag{A34}$$

$$\frac{dLIT_{s,C}}{dt} = I_{LIT_{s,C}} \times \left(1 - f_{i,struc}\right) - A2 - A6, \tag{A35}$$

$$\frac{dMIC_{r,C}}{dt} = CUE_{[1]} \times (A1 + A3) + CUE_{[2]} \times (A2) - A4 - Overflow_r, \tag{A36}$$

$$\frac{dMIC_{K,C}}{dt} = CUE_{[3]} \times (A5 + A7) + CUE_{[4]} \times (A6) - A8 - Overflow_K, \tag{A37}$$

$$\frac{dSOM_{p,C}}{dt} = I_{LIT_{m,C}} \times f_{i,met} + (f_{p,r} \times A4) + (f_{p,K} \times A8) - A9, \tag{A38}$$

$$\frac{dSOM_{c,C}}{dt} = I_{LIT_{s,C}} \times f_{i,struc} + (f_{c,r} \times A4) + (f_{c,K} \times A8) - A10, \tag{A39}$$

$$\frac{dSOM_{a,C}}{dt} = (f_{a,r} \times A4) + (f_{a,K} \times A8) + A9 + A10 - A3 - A7, \tag{A40}$$

$$\frac{dLIT_{m,N}}{dt} = \frac{I_{LIT_{m,C}} \times \left(1 - f_{i,met}\right)}{CN_m} - A11 - A15, \tag{A41}$$

$$\frac{dLIT_{s,N}}{dt} = \frac{I_{LIT_{s,C}} \times \left(1 - f_{i,struc}\right)}{CN_s} - A12 - A16, \tag{A42}$$

$$\frac{dMIC_{r,N}}{dt} = NUE \times (A11 + A13 + A12) - A14 + DINup_r - Nspill_r, \tag{A43}$$

$$\frac{dMIC_{K,N}}{dt} = NUE \times (A15 + A17 + A16) - A18 + DINup_K - Nspill_K, \tag{A44}$$

$$\frac{dSOM_{p,N}}{dt} = \frac{I_{LIT_{m,C}} \times \left(f_{i,met}\right)}{CN_m} + (f_{p,r} \times A14) + (f_{p,K} \times A18) - A19, \tag{A45}$$

$$\frac{dSOM_{c,N}}{dt} = \frac{I_{LIT_{s,C}} \times \left(f_{i,struc}\right)}{CN_s} + (f_{c,r} \times A14) + (f_{c,K} \times A18) - A20, \tag{A46}$$

$$\frac{dSOM_{a,N}}{dt} = (f_{a,r} \times A14) + (f_{a,K} \times A18) + A19 + A20 - A13 - A17, \tag{A47}$$

$$\frac{dDIN}{dt} = (1 - NUE) \times (A11 + A12 + A13 + A15 + A16 + A17) +$$
$$Nspill_r + Nspill_K - DINup_r - DINup_K - LeachingLoss. \tag{A48}$$

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





**750   Table 1. Parameters used in MIMICS-CN for both LIDET and equilibrium simulations.**

| Parameter | Description | Value | Units |
|---|---|---|---|
| $f_{met}$ | Partitioning of inputs to metabolic litter pool | 0.85 - 0.013 (lignin/N) | - |
| $f_i$ | Fraction of litter inputs transferred to SOM | 0.05, 0.3 | - |
| $V_{slope}$ (Met-r, Met-K, Struc-r) | Regression coefficient (Eq. 1) | 0.063 | ln(mg C (mg MIC)$^{-1}$ h$^{-1}$)°C$^{-1}$ |
| $V_{slope}$ (Struc-K, Avail-r, Avail-K) | Regression coefficient (Eq. 1) | 0.043 | ln(mg C (mg MIC)$^{-1}$ h$^{-1}$)°C$^{-1}$ |
| $V_{int}$ | Regression intercept (Eq. 1) | 5.47 | ln(mg C (mg MIC)$^{-1}$ h$^{-1}$) |
| $a_V$ | Tuning coefficient (Eq. 1) | $4.8 \times 10^{-7}$ | - |
| $V_{mod}$ | Modifies $V_{max}$ | 10, 1.5, 10, 3, 2.25, 2 | - |
| $P_{scalar}$ | Physical protection scalar used in $K_{mod}$ | $(2 \times e^{-2 \times \sqrt{(fclay)}})^{-1}$ | - |
| $K_{slope}$ (Met-r, Met-K, Avail-r, Avail-K) | Regression coefficient (Eq. 2) | 0.017 | ln(mg C cm$^{-3}$)°C$^{-1}$ |
| $K_{slope}$ (Struc-r, Struc-K) | Regression coefficient (Eq. 2) | 0.027 | ln(mg C cm$^{-3}$)°C$^{-1}$ |
| $K_{int}$ | Regression intercept (Eq. 2) | 3.19 | ln(mg C cm$^{-3}$) |
| $a_K$ | Tuning coefficient (Eq. 2) | 0.5 | - |
| $K_{mod}$ | Modifies $K_m$ | 0.125, 0.5, 0.25 × $P_{scalar}$, 0.5, 0.25, 0.167 × $P_{scalar}$ | - |
| KO | Further modifies $K_m$ for oxidation of SOM$_c$ | 6, 6 | - |
| τ | Microbial biomass turnover rate | $2.4 \times 10^{-4} \times e^{0.3\,(fmet)} \times \tau_{mod1} \times \tau_{mod2}$, $1.1 \times 10^{-4} \times e^{0.1\,(fmet)} \times \tau_{mod1} \times \tau_{mod2}$ | h$^{-1}$ |
| $\tau_{mod1}$ | Modifies microbial turnover rate | 0.6 < √(NPP/100) < 1.3 | - |
| $\tau_{mod2}$ | Modifies microbial turnover rate | τ × 0.55 / (.45 × Inputs) | - |
| β | Exponent that modifies turnover rate | 2 | - |
| CUE | Microbial carbon use efficiency | 0.55, 0.25, 0.75, 0.35 | mg mg$^{-1}$ |
| NUE | Proportion of mineralized N captured by microbes | 0.85 | mg mg$^{-1}$ |
| CN$_s$ | C:N of structural litter | (Measured CN − CN$_m$ × $f_{met}$) / (1- $f_{met}$) | mg mg$^{-1}$ |
| CN$_m$ | C:N of metabolic litter | 15 | mg mg$^{-1}$ |
| CN$_r$ | C:N of copiotrophic microbial pool | 6 | mg mg$^{-1}$ |
| CN$_k$ | C:N of oligotrophic microbial pool | 10 | mg mg$^{-1}$ |
| $f_p$ | Fraction of τ partitioned to SOM$_p$ | $0.015 \times e^{1.3\,(fclay)}$, $0.01 \times e^{0.8\,(fclay)}$ | - |
| $f_c$ | Fraction of τ partitioned to SOM$_c$ | $0.3 \times e^{-3\,(fmet)}$, $0.9 \times e^{-3\,(fmet)}$ | - |
| $f_a$ | Fraction of τ partitioned to SOM$_a$ | 1 - ($f_p + f_c$) | - |
| D | Desorption rate from SOM$_p$ to SOM$_a$ | $10^{-6} \times e^{-4.5\,(fclay)}$ | h$^{-1}$ |
| $N_{leak}$ | Rate of loss of inorganic N pool | 0.2 | h$^{-1}$ |



**Table 2. Goodness-of-fit statistics comparing MIMICS-CN and DAYCENT simulations to observations of C and N in decomposing litterbags in the LIDET study, aggregated by biome. DAYCENT results are subset from simulations in Bonan et al. (2013) to match the sites included in MIMICS-CN simulations. The values shown are the number of observations (n), Pearson's correlation coefficient squared ($R^2$), root mean square error (RMSE), and bias calculated between observed and simulated percent C and N remaining. For more details on the sites grouped into each biome, see Wieder et al. (2015).**

| Biome | n | MIMICS-CN Carbon $R^2$ | RMSE | bias | DAYCENT Carbon $R^2$ | RMSE | bias | MIMICS-CN Nitrogen $R^2$ | RMSE | bias | DAYCENT Nitrogen $R^2$ | RMSE | bias |
|---|---|---|---|---|---|---|---|---|---|---|---|---|---|
| Tundra | 114 | 0.74 | 12.56 | 9.49 | 0.78 | 8.32 | 3.21 | 0.33 | 0.32 | 0.09 | 0.41 | 0.31 | 0.00 |
| Boreal | 60 | 0.61 | 14.30 | 9.32 | 0.73 | 9.06 | -0.55 | 0.64 | 0.28 | 0.07 | 0.72 | 0.27 | -0.14 |
| Conifer | 60 | 0.79 | 18.61 | -16.42 | 0.89 | 9.09 | 5.93 | 0.73 | 0.20 | 0.05 | 0.79 | 0.26 | 0.13 |
| Deciduous | 94 | 0.59 | 16.40 | -8.92 | 0.80 | 12.36 | 9.20 | 0.51 | 0.31 | -0.13 | 0.63 | 0.33 | 0.18 |
| Humid | 151 | 0.50 | 17.24 | -3.23 | 0.61 | 15.18 | -4.22 | 0.14 | 0.44 | -0.13 | 0.24 | 0.45 | -0.04 |
| Arid | 113 | 0.61 | 16.67 | 2.09 | 0.68 | 19.90 | 11.63 | 0.32 | 0.29 | 0.16 | 0.01 | 0.49 | 0.20 |
| Tropical | 46 | 0.57 | 15.29 | 7.75 | 0.64 | 20.81 | 17.04 | 0.46 | 0.45 | 0.36 | 0.20 | 0.55 | 0.35 |
| All | 638 | 0.63 | 16.00 | -0.12 | 0.67 | 14.36 | 4.73 | 0.29 | 0.34 | 0.03 | 0.30 | 0.40 | 0.08 |









**Table 3. Ranges of MIMICS-CN estimates of steady-state values for a variety of soil pools and fluxes, compared against observed ranges from several continent-wide data synthesis studies. The ranges of values included for MIMICS-CN are derived from simulations of sites included in the LIDET study.**

| | MIMICS-CN range | Published range | Reference |
|---|---|---|---|
| Total C (mg cm$^{-3}$)* | 7.0-50 | 3.9-89 | Zak et al. 1994 |
| | | 2.7-360 | Xu, Thornton and Post 2013 |
| | | 5.2-610 | Cleveland and Liptzin 2007 |
| Total N (mg cm$^{-3}$)* | 0.60-5.1 | 0.38-5.1 | Zak et al. 1994 |
| | | 0.66-22 | Xu, Thornton and Post 2013 |
| | | 0.39-24 | Cleveland and Liptzin 2007 |
| Soil C:N | 9.6-12 | 4.0-40 | Colman and Schimel 2013 |
| | | 10-28 | Zak et al. 1994 |
| | | 11-31 | Xu, Thornton and Post 2013 |
| | | 2.0-82 | Cleveland and Liptzin 2007 |
| Inorganic nitrogen (µg cm$^{-3}$) | 0.01-0.06 | 0.12-8.1 | Zak et al. 1994 |
| Respiration (µg C cm$^{-3}$ hr$^{-1}$) | 0.02-0.28 | 0.01-0.70 | Colman and Schimel 2013 |
| | | 0.21-0.91 | Zak et al. 1994 |
| Net N mineralization (µg N cm$^{-3}$ hr$^{-1}$) | 0-0.01 | 0-0.10 | Colman and Schimel 2013 |
| | | 0.004-0.058 | Zak et al. 1994 |
| Microbial biomass C (mg cm$^{-3}$) | 0.15-1.3 | 0.03-1.3 | Zak et al. 1994 |
| | | 0.01-5.3 | Xu, Thornton and Post 2013 |
| | | 0.08-39 | Cleveland and Liptzin 2007 |
| Microbial biomass N (mg cm$^{-3}$) | 0.02-0.16 | 0.006-0.33 | Zak et al. 1994 |
| | | 0.042-0.64 | Xu, Thornton and Post 2013 |
| | | 0.018-4.9 | Cleveland and Liptzin 2007 |
| Microbial biomass C as % of soil C | 0.95-4.8 | 0.18-3.3 | Zak et al. 1994 |
| | | 0.99-5.0 | Xu, Thornton and Post 2013 |
| | | 0.27-93 | Cleveland and Liptzin 2007 |
| Microbial biomass N as % of soil N | 1.2-5.9 | 1.1-15 | Zak et al. 1994 |
| | | 2.3-5.7 | Xu, Thornton and Post 2013 |
| | | 0.48-64 | Cleveland and Liptzin 2007 |

*Depths simulated by MIMICS-CN are for the top 30 cm of soil, whereas published ranges represent measurements ranging from the top 5 to top 30 cm.



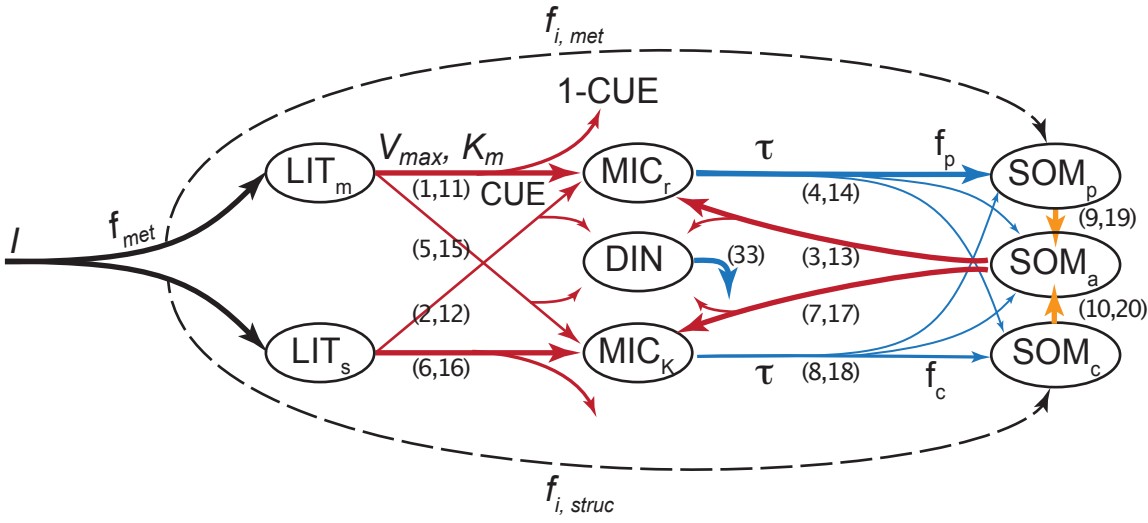

**Figure 1. Overview of the pools and fluxes of C and N in MIMICS-CN. Litter inputs (I) are determined based on site-specific net primary productivity and partitioned between metabolic and structural litter pools (LITm and LITs) using a site-specific litter quality metric ($f_{met}$) calculated using litter lignin and N content. Temperature-sensitive forward Michaelis-Menten kinetics ($V_{max}$ and $K_m$, red lines) determine the flux of litter pool C and N and available SOM C and N ($SOM_a$) into microbial biomass ($MIC_r$ and $MIC_K$). Fluxes of C into microbial pools result in respiration losses according to a defined carbon use efficiency (CUE). Microbes maintain biomass stoichiometry by spilling excess C as overflow respiration or excess N into the dissolved inorganic nitrogen pool (DIN) based on a prescribed biomass C:N. Microbial biomass turnover ($\tau$, blue) varies by functional type ($MIC_r$ and $MIC_K$) and is proportional to the square of microbial biomass. Microbial biomass turns over into available ($SOM_a$), physicochemically-stabilized ($SOM_p$) and chemically-stabilized ($SOM_c$) soil organic matter pools. Inorganic N (DIN) leaks from the model at a first-order rate. Numbers in parentheses indicate the equations in Appendix A that correspond to each depicted flux. Parameter values, units and descriptions are given in Table 1.**



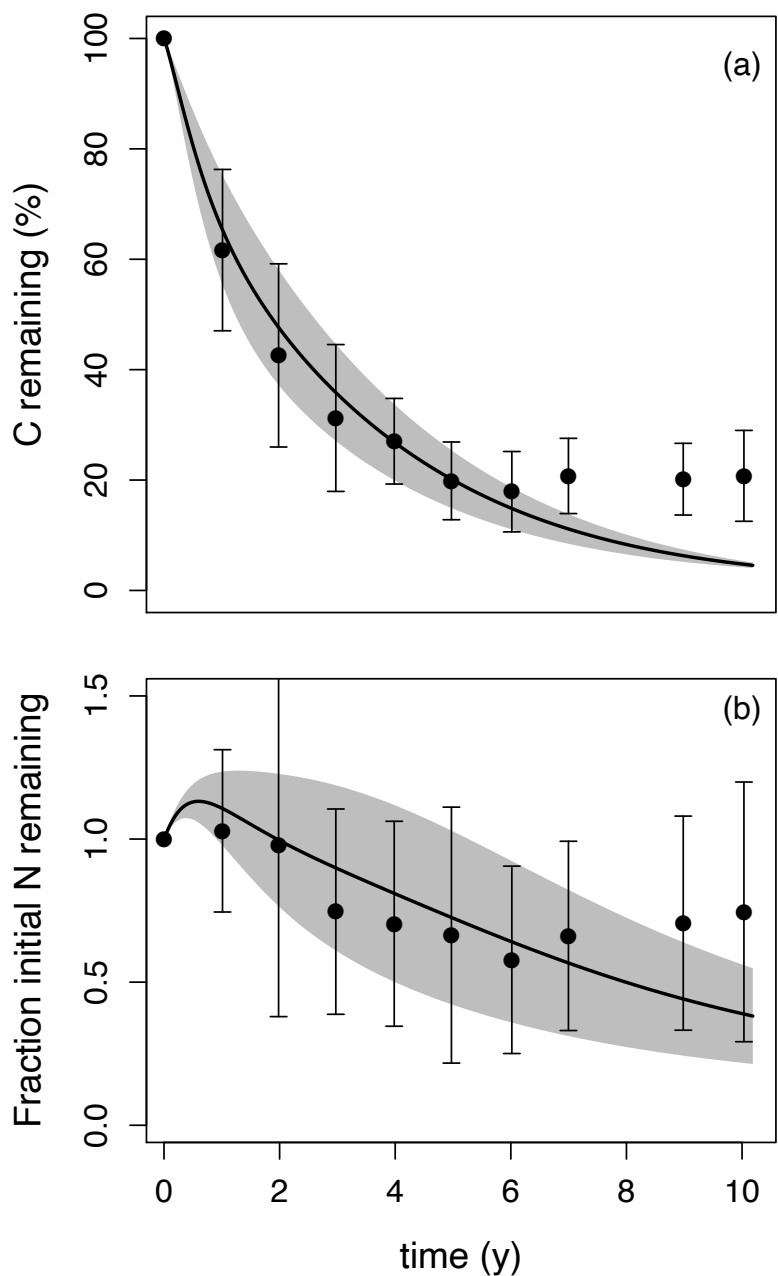

**Figure 2. Litter decomposition timeseries simulated by MIMICS-CN (lines with shaded area) compared to observations (points and error bars) of (a) percent mass remaining and (b) fraction of N remaining over ten years for six different litter types at the Harvard Forest LTER. Litter decomposition data came from the LIDET study (Parton et al., 2007; Bonan et al., 2013; mean ±1 SD). Spread in the observations and model are largely generated by the effects of initial litter quality on decomposition rates and N dynamics. Model parameters were calibrated to fit MIMICS-CN to observations from Harvard Forest (Table 1).**





**Figure 3. MIMICS-CN simulations of percent C remaining (top) and N remaining (bottom) in litterbags in the LIDET study versus observed values, colored by litter type (left) or biome (right). Dashed line shows the 1:1 line.**



**Figure 4. MIMICS-CN simulations of immobilization-mineralization thresholds across litters of different quality. Litter quality (in terms of C:N and lignin content) decreases from upper left panel to lower right panel. Red dots show model simulations of C losses vs N losses from litterbags in the LIDET study. Colored dots show observed C vs N losses across biomes (Parton et al. 2007).**





815

**Figure 5. Distributions of MIMICS-CN estimates of steady-state values for a variety of soil pools and fluxes, compared against observed ranges from several continent-wide data synthesis studies. Black lines show the median value across all observations; red lines show median value of MIMICS-CN simulations.**





**Figure 6. Variation in steady state SOM pools and environmental factors controlling their distribution in MIMICS-CN simulations across LIDET sites. Top panels show the (a) total C stocks in physicochemically-protected, chemically-protected, and available SOM pools (SOMp, SOMc, SOMa pools, respectively) arranged by the site mean annual temperature (MAT), or the (b) relative fraction of each SOM pool arranged in the same way. Upper right and bottom panels show the correlations between C in each SOM pool and environmental drivers including: (c) SOMp vs. the product of annual net primary productivity (ANPP) and clay content, (d) SOMc and SOMa vs. MAT, and (e) SOMc and SOMa vs. lignin content of litter inputs at each site. Finally, (f) soil stoichiometry is largely determined by the fraction of total SOM pools that are considered physicochemically protected.**