# Peer review of "Stoichiometrically coupled carbon and nitrogen cycling in the MIcrobial-MIneral Carbon Stabilization model version 1.0 (MIMICS-CN v1.0)"

_Geoscientific Model Development, 2019_

## Short Comment (SC1) · 7 Jan 2020

Dear authors,

in my role as Executive editor of GMD, I would like to bring to your attention our Editorial version 1.2:

https://www.geosci-model-dev.net/12/2215/2019/

This highlights some requirements of papers published in GMD, which is also available on the GMD website in the 'Manuscript Types' section:

http://www.geoscientific-model-development.net/submission/manuscript_types.html

[Figure]

In particular, please note that for your paper, the following requirement has not been met in the Discussions paper:

- "The main paper must give the model name and version number (or other unique identifier) in the title."

Please add a version number for MIMCS-CN in the title upon your revised submission to GMD.

Yours, Astrid Kerkweg
* * *

---

## Referee Comment (RC1) · Anonymous Referee #1 · 17 Jan 2020

This paper develops a new version of MIMICS with nitrogen cycling that is coupled to carbon using the standard theory of element limitation. Like its predecessor, MIMIC-CN is tested against litter decomposition experiments conducted across a wide range of sites, but this time focusing on C and N interactions. General fits of other pools and ratios are also shown to correspond with data collected from several synthesis studies. This paper is well within the scope of GMD, it describes the model clearly and provides a thorough continental-scale analysis of model predictions. I do not have any major issues with the paper but do have some suggestions that I hope will be useful.

It is interesting that initial litter quality does not control the soil C:N. I just

saw a paper that uses the bare fallow experiments to argue the opposite (https://www.nature.com/articles/s41598-019-55058-1), but with a very different model. Just something to think about – and possibly to address in the discussion.

L32: I suggest "These models *can be* as good as" instead of they are. You can say the models with microbes are better in the papers cited at the end of the sentence, but not about the untested models in the previous sentence (to which the subject "these models" seems to refer).

L206: MIMICS-CN does "as well as or better than DAYCENT" based on what criteria? From the following sentences, I assume and would make explicit here that you are using RMSE. Because using R2 I would say DAYCENT does a bit better, and using bias they are equivocal but in different ways (I don't think this undermines your development, since this model also provides uniquely testable predictions). I would also give a brief summary of the R2 and bias measures in text since they are in the table.

L220: I would change "red dots" to red triangles or symbols throughout, since the word dots makes me look for circles.

L273-274: This sentence sounds like you are guessing that microbes are C-limited, but you can know for sure by checking for overflow respiration.

L297: I agree that microbial community dynamics can add variability, but there are a lot of other possibilities also, like soil moisture and mineralogy (that which isn't captured by clay, like Fe and Al content).

L350: I think doing this is a reasonable way to get the right soil C:N given the structure of MIMICS, but I don't think POM/free light explains it all because that fraction is not physically protected. You could also be missing a protection mechanism that bypasses microbial biomass like aggregation of POM.

Figure 1: This figure is a little confusing because DIN is a N-only pool, but the other pools are C and N together. It might look a little unwieldy to show all C and N pools

[Figure]

since you have additional pools. You could shade/color the background or the outline of each pool with two colors to represent that it is partly C and partly N. Figure 1 of these papers try some different strategies: https://doi.org/10.5194/gmd-11-2111-2018, https://doi.org/10.1002/2017JG003796, https://doi.org/10.1016/j.soilbio.2009.02.031)

Figure 2 : Can these panels have the same y-axis ?

---

## Referee Comment (RC2) · Anonymous Referee #2 · 25 Jan 2020

General Comments:

This is a substantial effort to incorporate microbial controls into a modeling framework to simulate soil C and N dynamics over broad spatial and temporal scales. The MIMICS model follows the well-known CENTURY model structure used in many previous ecosystem and global scale models. It incorporates the CENTURY litter allocation scheme, but identifies two microbial guilds that each primarily processes one of the two litter pools, and allocates microbial products (and some litter) to three SOM pools reminiscent of- but more ecologically defined than the active, slow and passive pools in CENTURY. This model replaces first-order decay rates with Michaelis-Menten

functions based on microbial biomass, to calculate principle material fluxes, with kinetic parameters representing composite, empirical attributes of microbial pools. Thus decay rates are regulated by microbial biomass with stringent controls on microbial biomass constraining maximum rates. The results showed general correspondence with observations at least comparable to earlier results of more empirical models (e.g., DAYCENT) but generating potential insights (or at least insightful questions) to underlying microbial controls. The authors state (line 401) that MIMICS-CN is a "first step towards representing" a more comprehensive and realistic soil biogeochemical model, but I urge the authors to be bolder. In fact, there is little need to make so many comparisons to DAYCENT other than demonstrating the capacity of a more modern, biologically defined model to produce more insights. That argument seems to be generally accepted.

Specific Comments:

The influence of the CENTURY formulation on MIMIC's structure is apparent, but is there a rationale independent of convenience or of comparing behaviors of similarly structured models having different functional equations? For example, what is the justification for the two particular pools of microorganisms? Is it an extension of the litter quality definition?

What is the rationale for the three pools of SOM? This allocation isn't entirely consistent with other comparable models, like MEND, Millenial Model, etc. Although described in other papers, the central importance of this SOM scheme to MIMIC's behavior requires more explanation to understand simulations. Adding the dichotomy of physiochemical (mineral associated) and chemical (recalcitrance) is a positive step, but it seems to describe SOM as particulates and adsorbed organics without mention of soil aggregates that are important and tend to mix different qualities of dead organic matter. Is the active pool comprised of dissolved compounds? In addition, what empirically observed data were used for comparisons (e.g., Fig. 6)? Were these various fractions of soil extractions?

A novel aspect of this model is the dual "spilling" mechanisms for C and N, depending on the balance of supply vs. demand of C and N between substrate and microorganisms, and based on reasonable stoichiometric constraints. However, could this mechanism contribute to the excess loss of C (and N) over the long term? Also, it seems that the pulse of litter for the 10-year simulation represented the sole input for those years, correct? Could this also be a reason why simulated soil C and N were lower than observations at 7-10 years? Finally, what is the justification for nominal N-leakage (i.e., NUE = 0.85) and the N-leaching rate (eq. A33)? Could this N-loss be another factor contributing to lower C and N at late stages of simulation? I think the model has several features that could explain that discrepancy.

Why did the authors choose only 6 of the possible LIDET litter types, and these 6 in particular?

Lines 138-9: Was microbial biomass at Harvard derived from Xu et al. (2013) or observations?

Line 173: Not all previous SOM models simply used cascading pools of progressively more recalcitrant materials. The value of some of these final explorations isn't clear.

Line 188: How much of the similarity between microbial biomass estimates and observations (Fig. 5) can be ascribed to the density-sensitive turnover rate for microorganisms? Overall, the microbial values were the most tightly constrained within the model to reflect studies similar to those included within comparisons, especially as percent of soil C and N; if so, lines 235-237 might be overstated.

Lines 193-5: What was the rationale for the changes in (fi) and microbial turnover parameters other than to fit observations at Harvard Forest? Is there an interpretation of these adjustments?

Line 314 (and elsewhere): The authors seem surprised that MIMICS can reasonably match basic characteristics of these systems, but not only is the model largely constructed parallel to previous models that already did so, but additional model parameters and flexibility has been incorporated (and constrained). It would be a surprise if it didn't. Again, the authors could speak more boldly about their work.

Lines 353+: How do these pools compare to the particulate, aggregate protected, and mineral-associated organic matter pools that more realistically represent SOM (cf. Abramoff et al. 2017)? I don't understand how the pools defined in MIMICS were compared to observations.

Line 358: How do the first-order kinetics of the physiochemically-protected SOM compare with adsorption-desorption kinetics of mineral-associated organic matter (cf. Wang et al. 2013 Ecol Appl 23:255-272)?

Lines 354+: Soil clay content was important in MIMICS and obviously in the real world. This is a mechanism needed for broad scale modeling, but how does MIMICS' responsiveness differ from earlier models that explicitly included soil texture as a control on SOM pool dynamics?

Line 370: Wouldn't the relationship between soil C:N and litter C:N be strongly influenced by soil mineralogy and chemistry? Not that microbial processing wouldn't be important, but stabilization is likely impacted by the nature of the stabilizing medium.

Technical Suggestions:

Fig. 3: It seems that the individual R2 values for C and N by ecosystem in Table 2 represent the scatterplots in Fig. 3b and d, so it would be helpful to mention the biases reported in Table 2 when interpreting differences between simulations and observations by biome.

Are the simulation outputs in Fig. 4 red triangles rather than the dots mentioned in the legend? Also, I don't recall how the mass of N in decaying litter could increase above initial values; was this a result of immobilization from the soil DIN pool?

I don't think that section 4.3 adds much to the paper. If necessary, it could be tightened

to focus on the subset of topics that are the immediate objectives for future work by this group. Otherwise, it is so broad that it distracts from the important results of this work.

I suggest that most of lines 424-430 and 433-end could be omitted and the authors focus more explicitly on the key contributions of MIMICS-CN's to modeling soil C & N dynamics across broad scales. Again, I think the rest detracts from the interesting results of this work.

---

## Author Comment (AC1) · 21 Mar 2020

We thank Dr. Kerkweg for bringing our attention to the missing version number in the title of this manuscript, and will change the model descriptor in the title to "MIMICS-CN v1.0" to meet this requirement.

---

## Author Comment (AC2) · 1 Jun 2020

*Anonymous Referee #1:*

*1.1 "This paper develops a new version of MIMICS with nitrogen cycling that is coupled to carbon using the standard theory of element limitation. Like its predecessor, MIMICCN is tested against litter decomposition experiments conducted across a wide range of sites, but this time focusing on C and N interactions. General fits of other pools and ratios are also shown to correspond with data collected from several synthesis studies. This paper is well within the scope of GMD, it describes the model clearly and provides a thorough continental-scale analysis of model predictions. I do not have any major issues with the paper but do have some suggestions that I hope will be useful."*

**We appreciate the supportive and helpful comments from Referee #1. We agree with the bulk of their suggestions and have made modifications to the text and figures that we think addresses those suggestions.**

*1.2 "It is interesting that initial litter quality does not control the soil C:N. I just saw a paper that uses the bare fallow experiments to argue the opposite (https://www.nature.com/articles/s41598-019-55058-1), but with a very different model. Just something to think about – and possibly to address in the discussion."*

**This is an insightful comparison. We added text to the discussion to note that our "result directly contradicts a recent study using a first-order linear model which presumed that litter quality and soil quality at equilibrium were directly proportional (Menichetti et al., 2019). Although many soil biogeochemical models prescribe soil C:N ratios for individual pools, the stoichiometry of SOM in MIMICS-CN is an emergent property of the model." We also note that the current parameterization of MIMICS-CN provides predicts relatively low soil C:N ratios with little variation among sites (Fig. 5, Table 3).**

*1.3 "L32: I suggest "These models \*can be\* as good as" instead of they are. You can say the models with microbes are better in the papers cited at the end of the sentence, but not about the untested models in the previous sentence (to which the subject "these models" seems to refer)."*

**We agree; the text now reads "While these models serve different purposes, some can be as good as or better than models without explicit microbial pools…"**

*1.4 "L206: MIMICS-CN does "as well as or better than DAYCENT" based on what criteria? From the following sentences, I assume and would make explicit here that you are using RMSE. Because using R2 I would say DAYCENT does a bit better, and using bias they are equivocal but in different ways (I don't think this undermines your development, since this model also provides uniquely testable predictions). I would also give a brief summary of the R2 and bias measures in text since they are in the table."*

**Thank you for this feedback; the text now reads: "Across a broad range of biomes, MIMICS-CN and DAYCENT both show good agreement with LIDET observations. Across sites MIMICS-CN has similar $R^2$ and RMSE values but lower bias compared to DAYCENT for mass loss (MIMICS-CN: $R^2$=0.63, RMSE=16.0, bias=-0.12; DAYCENT: $R^2$ = 0.67, RMSE=14.4, bias=4.73), and percent N remaining ($R^2$=0.29, RMSE=0.34, bias=0.03; DAYCENT: $R^2$=0.30, RMSE=0.40, bias=0.08). Broadly, MIMICS-CN outperformed DAYCENT in the warmest biomes while DAYCENT excelled for colder sites for both C**

**and N (Table 2), but the differences in model fit to data were slight and would be difficult to attribute to any particular differences in model structure.”**

*1.5 "L220: I would change "red dots" to red triangles or symbols throughout, since the word dots makes me look for circles."*
**Changed; thank you.**

*1.6 "L273-274: This sentence sounds like you are guessing that microbes are C-limited, but you can know for sure by checking for overflow respiration."*
**We clarified the language to reference our specific simulations and model output values for overflow respiration. We did check for overflow respiration, but we agree the sentence was ambiguous. The text now reads "At equilibrium, microbes in our MIMICS-CN simulations primarily obtained N through recycling of SOM pools with favorably low C:N ratios, with the result that modeled microbes were almost always C-limited at equilibrium and rarely exhibited overflow respiration."**

*1.7 "L297: I agree that microbial community dynamics can add variability, but there are a lot of other possibilities also, like soil moisture and mineralogy (that which isn't captured by clay, like Fe and Al content)."*
**We agree and added some new language and relevant citations. The text now reads "Spatial variability in ecosystem processes, like N mineralization rates, may be linked to factors like local-scale microbial community composition, soil moisture, or mineralogy (Graham et al., 2016; Smithwick et al., 2005; Soranno et al., 2019; Doetterl et al., 2015)."**

*1.8 "L350: I think doing this is a reasonable way to get the right soil C:N given the structure of MIMICS, but I don't think POM/free light explains it all because that fraction is not physically protected. You could also be missing a protection mechanism that bypasses microbial biomass like aggregation of POM."*
**We agree, but we think that aggregation is a stabilization mechanism that has proven incredibly difficult to parameterize and model, and that it is beyond the scope of what MIMICS-CN attempts to capture. In the last paragraph we clarify that "Future work could compare model formulations that take different approaches to microbial community and stoichiometric parameters (e.g. flexible microbial parameters like C:N or CUE, additional microbial groups, partitioning microbial metabolism into a greater number of pathways) and refinement of mechanisms that confer SOM persistence."**

*1.9 "Figure 1: This figure is a little confusing because DIN is a N-only pool, but the other pools are C and N together. It might look a little unwieldy to show all C and N pools since you have additional pools. You could shade/color the background or the outline of each pool with two colors to represent that it is partly C and partly N. Figure 1 of these papers try some different strategies: https://doi.org/10.5194/gmd-11-2111-2018, https://doi.org/10.1002/2017JG003796, https://doi.org/10.1016/j.soilbio.2009.02.031)"*
**Thanks for this suggestion; we have revised Figure 1 to illustrate C and N pools in the model.**

*1.10    "Figure 2 : Can these panels have the same y-axis ?"*
**Done, changed axes to be more easily compared.**

---

## Author Comment (AC3) · 1 Jun 2020

*Anonymous Referee #2:*

*General Comments:*

*2.1 This is a substantial effort to incorporate microbial controls into a modeling framework to simulate soil C and N dynamics over broad spatial and temporal scales. The MIMICS model follows the well-known CENTURY model structure used in many previous ecosystem and global scale models. It incorporates the CENTURY litter allocation scheme, but identifies two microbial guilds that each primarily processes one of the two litter pools, and allocates microbial products (and some litter) to three SOM pools reminiscent of- but more ecologically defined than the active, slow and passive pools in CENTURY. This model replaces first-order decay rates with Michaelis-Menten paper functions based on microbial biomass, to calculate principle material fluxes, with kinetic parameters representing composite, empirical attributes of microbial pools. Thus decay rates are regulated by microbial biomass with stringent controls on microbial biomass constraining maximum rates. The results showed general correspondence with observations at least comparable to earlier results of more empirical models (e.g., DAYCENT) but generating potential insights (or at least insightful questions) to underlying microbial controls. The authors state (line 401) that MIMICS-CN is a "first step towards representing" a more comprehensive and realistic soil biogeochemical model, but I urge the authors to be bolder. In fact, there is little need to make so many comparisons to DAYCENT other than demonstrating the capacity of a more modern, biologically defined model to produce more insights. That argument seems to be generally accepted.*

**We appreciate these general comments and the reviewer's clear understanding and appreciation of the work presented here. We are grateful for the encouragement to be "bolder," and have made substantial changes to the discussion that we hope will address this feedback (see especially section 4.3 to the end). We still find the DAYCENT comparison helpful in contextualizing the parameterization and evaluation of MIMICS results, but will take care to avoid unnecessary or lengthy comparison in the discussion of the text.**

*Specific Comments:*

*2.2 The influence of the CENTURY formulation on MIMIC's structure is apparent, but is there a rationale independent of convenience or of comparing behaviors of similarly structured models having different functional equations? For example, what is the justification for the two particular pools of microorganisms? Is it an extension of the litter quality definition?*

**We clarified the intent and assumptions of the microbial functional groups represented in MIMICS-CN in the methods. The microbial functional groups are intended to broadly capture tradeoffs in microbial growth rates and growth efficiency, with rapidly-growing – low efficiency, r-strategist ($MIC_r$) and slower-growing – higher efficiency K-strategist ($MIC_K$; Wieder et al. 2015). In MIMICS-CN we extend these microbial physiological traits to include microbial stoichiometry and assume that the higher metabolic capacity of $MIC_r$ also require more nitrogen and, thus a lower microbial biomass C:N ratio.**

*2.3 What is the rationale for the three pools of SOM? This allocation isn't entirely consistent with other comparable models, like MEND, Millenial Model, etc. Although described in other papers, the central importance of this SOM scheme to MIMIC's behavior requires more explanation to understand simulations. Adding the dichotomy of physiochemical (mineral associated) and chemical (recalcitrance) is a positive step, but it seems to describe SOM as particulates and adsorbed organics without mention of soil aggregates that are important and tend to mix different qualities of dead organic matter. Is the active pool comprised of dissolved compounds? In addition, what empirically observed data were used for comparisons (e.g., Fig. 6)? Were these various fractions of soil extractions?*

**We consider the $SOM_p$ pool to be largely derived of low C:N organic matter that is largely composed of microbial necromass that is adsorbed onto mineral surfaces (e.g. Mineral associated organic matter, MAOM; Grandy and Neff, 2008). By contrast, the low-quality $SOM_c$ pool consists of decomposed or partially decomposed litter that has more structural C compounds, such as lignin, and a higher C:N ratio (e.g. particulate organic matter, POM). Finally, the $SOM_a$ is the only SOM pool that is available for microbial decomposition; it contains a mixture of fresh microbial residues, products that are desorbed from the $SOM_p$ pool, as well as depolymerized organic matter from the $SOM_c$ pool. Under these assumptions we do not specifically consider soil aggregates, but we recognize their importance in maintaining organic matter persistence in soils. We added text to the first section of the methods to describe the structure and reasoning behind MIMICS-CN in more detail. Figure 6 only illustrates model assumptions, not observations across sites (clarified further in R2.11, below).**

*2.4 A novel aspect of this model is the dual "spilling" mechanisms for C and N, depending on the balance of supply vs. demand of C and N between substrate and microorganisms, and based on reasonable stoichiometric constraints. However, could this mechanism contribute to the excess loss of C (and N) over the long term? Also, it seems that the pulse of litter for the 10-year simulation represented the sole input for those years, correct? Could this also be a reason why simulated soil C and N were lower than observations at 7-10 years? Finally, what is the justification for nominal N-leakage (i.e., NUE = 0.85) and the N-leaching rate (eq. A33)? Could this N-loss be another factor contributing to lower C and N at late stages of simulation? I think the model has several features that could explain that discrepancy.*

**We appreciate these suggestions and assume they are focused on trying to understand why our results also show higher than observed rates of litter C mass loss in deciduous and coniferous forest (Figs 2a, 3b; Table 2). This suggests that the partitioning of plant detrital inputs into litter pools that are chemically defined works well for initial stages of litter decay, but may not consider the changes in substrate chemistry or microbial community succession that occur in later stages of decomposition that slow rates of mass loss (Berg, 2000; Bradford et al., 2017; Melillo et al., 1989). Models that implicitly represent microbial activity capture this phenomena by using a three pool structure (Adair et al., 2008), and future studies can consider how to more mechanistically understand interactions between initial litter quality, decomposer communities, climate, nutrient availability and late-stage litter decay rates (e.g. Craine et al., 2007; Hobbie et al., 2012; Wickings et al., 2012) in models like MIMICS-CN.**

*2.5 Why did the authors choose only 6 of the possible LIDET litter types, and these 6 in particular?*

**We focus our analysis on six leaf litters that were simulated across all sites that have been used previously to evaluate litter decomposition dynamics in terrestrial models (Bonan et al. 2013, Parton et al. 2007, Wieder et al. 2015). Root litter types included in the original LIDET experiment were not included.**

*2.6 Lines 138-9: Was microbial biomass at Harvard derived from Xu et al. (2013) or observations?*

**The value we used as a target for microbial biomass was estimated at 1% of soil C based on Xu et al. (2013).**

*2.7 Line 173: Not all previous SOM models simply used cascading pools of progressively more recalcitrant materials. The value of some of these final explorations isn't clear.*

**We agree this sentence was distracting and have removed it from the text.**

*2.8 Line 188: How much of the similarity between microbial biomass estimates and observations (Fig. 5) can be ascribed to the density-sensitive turnover rate for microorganisms? Overall, the microbial values were the most tightly constrained within the model to reflect studies similar to those included within comparisons, especially as percent of soil C and N; if so, lines 235-237 might be overstated.*

**This is a very astute observation. Among other parameters, the density-dependence turnover rate is an important control on microbial biomass that we used to parameterize the model for Harvard Forest prior to our simulations at other sites. We used the same value for density-dependent turnover for the rest of the LIDET sites and produced a range of values for microbial biomass that did reflect a similar range to observations (Fig. 5). In general, we think the indicated lines reflect an accurate reporting of our results. However, we understand the concern and changed the language to be less definitive.**

*2.9 Lines 193-5: What was the rationale for the changes in (fi) and microbial turnover parameters other than to fit observations at Harvard Forest? Is there an interpretation of these adjustments?*

**In section 2.2 of the methods, we discuss how we used observations at Harvard Forest to help parameterize the model before evaluating the model's performance at other sites. The purpose of these adjustments was to fit observations at Harvard Forest, with the expectation that making these adjustments would help the model to perform more realistically at other sites.**

*2.10 Line 314 (and elsewhere): The authors seem surprised that MIMICS can reasonably match basic characteristics of these systems, but not only is the model largely constructed parallel to previous models that already did so, but additional model parameters and flexibility has been incorporated (and constrained). It would be a surprise if it didn't. Again, the authors could speak more boldly about their work.*

**We appreciate the encouragement to speak more boldly about our work and recognize that MIMICS-CN simulates microbial stoichiometry, microbial growth and turnover, and microbially-mediated decomposition, rather than using prescribed values as in models that**

lack explicit representation of microbes. This increases the power of MIMICS-CN to explore the microbial and biogeochemical processes underpinning model predictions. Following these suggestions, we have made substantive changes to the text of the manuscript. We have added more discussion about Figure 6 and the implications of the patterns illustrated there, while refining other parts of the discussion to give caveats and limitations a less outsized impact in the manuscript relative to a discussion of the model structure, implications, and future directions.

*2.11 Lines 353+: How do these pools compare to the particulate, aggregate protected, and mineral-associated organic matter pools that more realistically represent SOM (cf. Abramoff et al. 2017)? I don't understand how the pools defined in MIMICS were compared to observations.*

Results in Fig. 6 are intended to illustrate patterns in model results that *could* be compared to observations, but we don't know of data available across environmental gradients that could be used to sufficiently evaluate these assumptions at this time, and we also feel that such an evaluation of all these variables would fall outside the scope of this manuscript. Nonetheless, this exercise provides an opportunity to explore how model-defined assumptions about pool stabilization mechanisms drive potential responses of SOM pools to environmental variables. For example, the chemically-protected and available SOM pools in MIMICS-CN turn over based on temperature-sensitive Michaelis-Menten kinetics and litter chemistry (the later controlling allocation to litter pools the relative abundance of microbial functional groups). Therefore, in our simulations, $SOM_C$ pools (analogous to light fraction or POM pools) were negatively correlated with MAT and positively correlated with litter lignin content (Fig. 6d, 6e). Turnover of the physicochemically-protected SOM pool, on the other hand, occurs via first-order kinetics with a rate constant modified by clay content, and the equilibrium values of this pool in MIMICS-CN are determined by inputs that largely come from microbial biomass and biomass turnover rates (Fig. 1). Therefore, the equilibrium values of simulated $SOM_p$ (analogous to heavy fraction or MAOM pools) were positively correlated with the product of ANPP and clay content (Fig. 6c). We added text to the discussion to clarify the purpose and interpretation of the results shown in Figure 6.

*2.12 Line 358: How do the first-order kinetics of the physiochemically-protected SOM compare with adsorption-desorption kinetics of mineral-associated organic matter (cf. Wang et al. 2013 Ecol Appl 23:255-272)?*

This is an interesting question that we would like to explore in the future, but addressing it in the text here falls outside the scope of the discussion about environmental controls over SOM pools in MIMICS.

*2.13 Lines 354+: Soil clay content was important in MIMICS and obviously in the real world. This is a mechanism needed for broad scale modeling, but how does MIMICS' responsiveness differ from earlier models that explicitly included soil texture as a control on SOM pool dynamics?*

We point to previously published work here. In global simulations with the carbon-only version of MIMICS, these assumptions result in MIMICS projecting longer turnover soil C

**times and larger soil C pool in the tropics than other models (Koven et al. 2017, Wieder et al. 2018) and a higher vulnerability of high latitude soil C stocks (Wieder et al. 2015; 2019).**

*2.14 Line 370: Wouldn't the relationship between soil C:N and litter C:N be strongly influenced by soil mineralogy and chemistry? Not that microbial processing wouldn't be important, but stabilization is likely impacted by the nature of the stabilizing medium.*
**We agree; the text now reads "is SOM stoichiometry correlated with litter quality, or is it better explained by climate, edaphic, and mineralogical gradients that impact soil microbial community composition, microbial activity, and mineral-mediated mechanisms of SOM persistence?"**

*Technical Suggestions:*
*2.15 Fig. 3: It seems that the individual R2 values for C and N by ecosystem in Table 2 represent the scatterplots in Fig. 3b and d, so it would be helpful to mention the biases reported in Table 2 when interpreting differences between simulations and observations by biome.*
**To avoid redundancies, we have removed the biome statistics from the text and refer to Table 2 (see also response to R1.4).**

*2.16 Are the simulation outputs in Fig. 4 red triangles rather than the dots mentioned in the legend? Also, I don't recall how the mass of N in decaying litter could increase above initial values; was this a result of immobilization from the soil DIN pool?*
**We changed "dots" to "triangles" in the legend and text, see also response to R1.5. The increase in N in litterbags above 100% of initial values was the result of immobilization from the soil DIN pool; we added a sentence to the methods to make this clearer.**

*2.17 I don't think that section 4.3 adds much to the paper. If necessary, it could be tightened to focus on the subset of topics that are the immediate objectives for future work by this group. Otherwise, it is so broad that it distracts from the important results of this work.*
**Following this suggestion, we edited the section referenced here (now 4.4) to give it a narrower focus and refine the broad discussion of potential next steps with MIMICS. Our goal with these changes was to highlight more specifically the next steps with MIMICS that we feel are the highest priority. We hope you find these changes clarified this portion of the discussion into something less distracting and more in line with the rest of the text.**

*2.18 I suggest that most of lines 424-430 and 433-end could be omitted and the authors focus more explicitly on the key contributions of MIMICS-CN's to modeling soil C & N dynamics across broad scales. Again, I think the rest detracts from the interesting results of this work.*
**We omitted most of the section referred to here and integrated it with the section above.**